

# The role of juvenile hormone in regulating reproductive physiology and dominance in *Dinoponera quadriceps* ants

Victoria C. Norman[1,2], Tobias Pamminger[2], Fabio Nascimento[3] and William O.H. Hughes[2]

[1] School of Biology, University of Leeds, Leeds, United Kingdom
[2] School of Life Sciences, University of Sussex, Brighton, United Kingdom
[3] Departamento de Biologia, Universidade de São Paulo, Ribeirão Preto, Brazil

Corresponding author
William O.H. Hughes,
william.hughes@sussex.ac.uk

## ABSTRACT

Unequal reproductive output among members of the same sex (reproductive skew) is a common phenomenon in a wide range of communally breeding animals. In such species, reproductive dominance is often acquired during antagonistic interactions between group members that establish a reproductive hierarchy in which only a few individuals reproduce. Rank-specific syndromes of behavioural and physiological traits characterize such hierarchies, but how antagonistic behavioural interactions translate into stable rank-specific syndromes remains poorly understood. The pleiotropic nature of hormones makes them prime candidates for generating such syndromes as they physiologically integrate environmental (social) information, and often affect reproduction and behaviour simultaneously. Juvenile hormone (JH) is one of several hormones that occupy such a central regulatory role in insects and has been suggested to regulate reproductive hierarchies in a wide range of social insects including ants. Here we use experimental manipulation to investigate the effect of JH levels on reproductive physiology and social dominance in high-ranked workers of the eusocial ant *Dinoponera quadriceps*, a species that has secondarily reverted to queenless, simple societies. We show that JH regulated reproductive physiology, with ants in which JH levels were experimentally elevated having more regressed ovaries. In contrast, we found no evidence of JH levels affecting dominance in social interactions. This could indicate that JH and ovary development are decoupled from dominance in this species, however only high-ranked workers were investigated. The results therefore confirm that the regulatory role of JH in reproductive physiology in this ant species is in keeping with its highly eusocial ancestors rather than its secondary reversion to simple societies, but more investigation is needed to disentangle the relationships between hormones, behaviour and hierarchies.

## INTRODUCTION

In many group-living animals, reproduction is not equally distributed among the breeding members. This phenomenon, known as reproductive skew, occurs in a wide range of communally-breeding species including birds, fishes and insects (*Jamieson, 1997*; *Cuvillier-Hot et al., 2004*; *Neff, Pitcher & Ramnarine, 2008*). Due to its fundamental implications for both ecological and evolutionary processes, this topic has attracted attention over the past decades from both a theoretical (see *Kokko & Johnstone, 1999*; *Johnstone, 2000*; *Kokko, 2003*) and empirical perspective in a variety of study systems (*Bourke, Green & Bruford, 1997*; *Field et al., 1998*; *Reeve & Keller, 2001*; *Widdig et al., 2004*).

Social insects, in particular ants, have emerged as important model systems to test some of the main predictions of reproductive skew theory due to their wide range of social complexity and life history strategies (*Reeve & Keller, 2001*). The majority of ant species are highly eusocial, with complex societies in which one or several queens produce all female offspring and her unmated daughter workers perform all other tasks such as brood care, foraging and nest defence (*Hölldobler & Wilson, 1990*). In such systems, as a result of the haplodiploid sex determination in Hymenoptera, workers are only able to produce male offspring and conflicts between queens and workers, or between workers, arise over male parentage (*Ratnieks & Reeve, 1992*; *Ratnieks, Foster & Wenseleers, 2006*). This group conflict and how it is resolved has generated a plethora of ground-breaking work, revolutionizing our understanding of group formation, conflict and maintenance (*Ratnieks, Foster & Wenseleers, 2006*).

While most modern ants have a specialized queen caste, some genera, such as *Dinoponera*, have secondarily reverted to simple, queenless societies in which reproduction is monopolized instead by mated, reproductively active workers called gamergates (*Peeters, 1997*). In most gamergate systems, all workers have the potential to become the dominant reproductive, resulting in strong within-group conflict over reproduction (*Peeters, 1997*). Such conflicts are often resolved via aggressive behavioural interactions that establish a dominance hierarchy in which only a single, or a small group of workers go on to reproduce. The question of how such ritualized physical aggression is physiologically translated into stable reproductive hierarchies with lower ranked workers not only remaining reproductively inactive but also assuming helper roles, remains poorly understood. In some species such as *D. quadriceps*, subordinate workers play a role in stabilizing the dominance hierarchy (*Monnin & Peeters, 1999*; *Monnin et al., 2002*).

Hormones are prime candidates for the proximate mechanisms underlying this process, because they not only physiologically integrate social stimuli including stress, but also regulate numerous other essential processes in adult insects such as reproduction, maternal behaviour and aggression (*Nijhout, 1998*; *Sasaki, Yamasaki & Nagao, 2007*; *Tibbetts & Huang, 2010*). There are several hormones that have been implicated in this, including dopamine, ecdysone and vitellogenin, with juvenile hormone (JH) being perhaps the best studied. However, it appears that JH can have contrasting effects in different taxa. In primitively eusocial species, such as paper wasps and bumblebees, JH is gonadotropic, stimulating ovary development in the same way as in solitary insects and resulting in

individuals being more socially dominant, while in the highly eusocial honeybee, in contrast, JH has lost its gonadotropic effect and instead is involved in regulating division of labour (*Robinson & Vargo, 1997*). In highly eusocial *Lasius niger* ants, the gonadotropic effect of JH has not only been lost but reversed, with higher JH levels being associated with reduced egg production (*Pamminger et al., 2016*; *Pamminger, Treanor & Hughes, 2016*). In addition JH has been shown to trigger foraging behaviour in some ants, making it a possible candidate to coordinate not only reproductive division of labour, but also division of labour between workers (*Robinson & Vargo, 1997*; *Norman & Hughes, 2016*). This might suggest a relatively simple switch in the action of JH with the evolution of complex eusocial societies, but it appears that the evolution dynamics of JH mode of action is more complex than that. In *Solenopsis* and *Pogonomyrmex*, ant genera with complex societies, JH exhibits stimulatory functions during reproduction (*Brent & Vargo, 2003*; *Libbrecht et al., 2013*). In *Streblognathus* and *Diacamma*, ants with simple, queenless societies, low JH titres in gamergates correlates with high individual ranks within the hierarchy and JH application will result in a loss of the reproductive status of the alpha (*Sommer, Hölldobler & Rembold, 1993*; *Cuvillier-Hot, Lenoir & Peeters, 2004*; *Brent et al., 2006*). In *Harpegnathos* ants, which also have simple, societies, where gamergates can reproduce following the founding queen's death, JH levels do not differ between reproductive and non-reproductive individuals, and experimental elevation of JH levels through the application of Juvenile Hormone analogue had no effect on egg production (*Penick, Liebig & Brent, 2011*).

Here we investigate the effects of JH on reproductive physiology and social dominance in the queenless ponerine ant *Dinoponera quadriceps*, by using topical application of the JH analogue (JHa) methoprene to experimentally manipulate JH levels. This species is of particular interest because it is one of ca. 100 species to have undergone an evolutionary reversion from a highly eusocial ancestor with a queen caste back to its basal state with queenless, simple societies (*Peeters, 1997*; *Monnin & Peeters, 1998*). All females in *D. quadriceps* are morphologically identical, with a single dominant gamergate, the alpha, actively suppressing a group of the higher ranked workers from becoming reproductively active with ritualized physical aggression including antennal blocking and boxing (*Monnin & Peeters, 1998*; *Grainger et al., 2014*). The presence of an alpha within the colony not only inhibits ovary activation in workers, the first step towards becoming reproductively active, but also results in submissive behaviour by subordinates (*Smith et al., 2011*; *Asher et al., 2013*). The physiological phenotypic differences between alphas and subordinates result from subtle differences in transcriptional network organisation, involving both conserved and novel genes (*Patalano et al., 2015*). If JH functions similarly to other queenless ant species, then we predict JH will suppress ovarian development and cause high ranking workers to decrease in status. If JH functions as a gonadotropin, similar to solitary insects and social wasps, then we predict JH to activate ovarian development and potentially to move up in the hierarchy. We focus for our experimental manipulation on high-ranked, but not alpha, workers because these have both the potential to move up the hierarchy to become reproductives and the potential to lose their position in the social hierarchy and become middle or low-ranked workers. We measure the effect of JH manipulation on ovary development and dominance behaviour. If JH links reproduction and hierarchy-related

behaviours it would then provide a proximate physiological explanation for rank-associated trait syndromes.

## METHODS

We used 13 colonies of *D. quadriceps,* which were collected from Bahia state, Brazil in November 2014 under permit from Instituto Brasileiro do Meio Ambiente e dos Recursos Naturais (IBAMA; 14BR004553). All colonies were maintained in the lab at 27 °C and 80% relative humidity for at least six months before the experiment. Colonies were fed with *Tenebrio molitor* larvae and apple, and had *ad libitum* access to water. Each individual was uniquely marked on the pronotum with numbered tags.

### Establishing the dominance hierarchy

Firstly, to establish the dominance hierarchy, colonies were monitored daily for two weeks, with the behaviours and locations of each individual being recorded once each day for 14 days. Observations lasted until each individual per colony had been recorded. Individuals showed high levels of consistency in behaviour and location during this period. Given the positive association in this species between an individual being of high rank and it interacting with brood (*Monnin & Peeters, 1999*; *Asher et al., 2013*), any individual that was observed at least once interacting with brood over the 14 day initial observation period was selected to undergo pairwise isolated dyadic interactions to narrow down their position in the social hierarchy. This method pairs every combination of ants sampled to observe which individual in each dyad is the dominant and which the subordinate, based on a characteristic dominance behaviour; this has previously been shown to be a reliable and robust way to establish dominance hierarchies in this species (*Grainger et al., 2014*). For this, individuals were taken from their colonies and placed individually in pots (85 mm × 75 mm × 55 mm) and allowed to acclimatise for 15 min. Pairs of ants were then placed in a new pot, their dominance interaction was observed and the dominant ant was recorded. This is indicated by only one behaviour in this context: dominant ants stand tall with their antennae either side of the subordinate individual which has antennae laid flat back behind their head (*Grainger et al., 2014*). This reaction normally occurs within the first 60 s of contact between pairs when it is expressed. We then ranked individuals based on the number of times they expressed dominance and assigned ranks to each individual. The higher-ranking individuals that ranked directly below the alpha and clearly above the remainder of the colony were then selected for the study (two or three high-ranking workers per colony, and 32 in total). Of these, 16 high-ranking workers were treated with a JH analogue (at least one per colony) and 16 as controls (at least one per colony; see below).

### Worker size and weight

Before the start of the experiment all selected workers were immobilized on ice for 1 min and their head width (maximal interorbital distance) measured as proxy for body size, as well as their fresh weight using a Precisa 125A balance.

## Behavioural measures and experimental procedure

Behavioural observations were made daily for five days before the first application of treatments to determine how consistent ants were for a number of behavioural variables. In total, observations took roughly 3–4 h per day, until each individual per colony had been recorded. We carried out daily scans for five days prior to treatment in which we recorded for each focal ant whether or not it was showing any aggression (either within the nest to conspecifics or gaping it's mandibles in defence outside of the nest), whether or not it was showing any brood care behaviours, and two measures of 'sociability': the distance to the nearest ant and the number of ants within 5 cm of the focal ant ('contacts'). During the same time interval we carried out individual-level assays for activity level, 'boldness' and defensive aggression, with the expectation that high rankers would show low activity level (*Monnin, Ratnieks & Brandão, 2003*), low boldness (as they are based inside and away from any 'risky' tasks such as nest defence or foraging (*Nascimento et al., 2012*; *Asher et al., 2013*)), and high levels of aggression (known to be associated with higher ranks (*Monnin & Peeters, 1999*; *Cant, Llop & Field, 2006*). General activity level was determined simply by placing the focal ant in a 90 mm Petri dish lined with filter paper, leaving it to acclimatise for 2 min, and then videoing the ant for 5 min using a Logitech c920 webcam. Speed of movement was quantified from videos using AntTrak path analysis software (*Tranter et al., 2014*). 'Boldness' was determined by placing the focal ant in a 90 mm Petri dish lined with filter paper and half blackened out with tape across the lid and sides, leaving it to acclimatise for 2 min, and then videoing the ant for 5 min to allow the proportion of time spent in the light half of the Petri dish to be calculated (less bold ants spend more time hiding in the darkened area of the Petri dish). Defensive aggression was determined by placing the focal ant in a pot (85 mm × 75 mm × 55 mm), leaving it to acclimatise for 5 min, and then tapping it gently on the head with the tip of a toothpick, as in *Pamminger et al. (2014)*. The reaction of the ant was ranked (0 = ignore, 1 = antennate, 2 = gape mandibles in a threat response, 3 = bite).

Following the initial assessment of individual behaviour, ants were assigned randomly to either the methoprene treatment or acetone solvent control (CoA) (with at least 1 methoprene treated and 1 control treated ant per colony), and all subsequent behavioural observations were conducted with the observer blind to treatment. For the methoprene treatment, a dose of 16.5 µg of methoprene (PESTANAL®; Sigma Aldrich, St. Louis, MO, USA) in 5 µl acetone was applied to the pronotum three times over a period of 1 week; control ants received 5 µl acetone on the same occasions. This dose was determined during a preliminary experiment and is low compared to the amounts used in other social insect studies (Table S1), indicating that the observed effects are not caused by potential toxic effects of JH at high doses. After two days of acclimatisation post-treatment, we repeated the behavioural observations. We carried out the assays daily for 4 days and on the 5th day carried out dyadic interaction assays between the focal ants and all other workers that had been observed performing brood care behaviour over the past three weeks. This enabled us to determine if the methoprene treatment had not only affected behaviour but also the position of the focal high rank ants in the hierarchy. Following the dyadic interactions, ants were freeze-killed in liquid nitrogen and stored at −80 °C until ovary dissection.

## Ovary dissection and fertility estimates

Ant ovaries were dissected under a Leica S8AP0 stereo microscope and the ovaries were transferred into Ringer solution. The ovaries were photographed using a Leica DFC 295 Camera and the Leica application suite software v. 4.1.0. Three ovarioles were randomly selected for further analysis to keep consistency between individuals. Using a Pyser-SGI® S78 stage micrometer 1.0/0.01 mm and the software ImageJ 1.47v, we measured the minimum, maximum and average width of the third of the ovarioles closest to the oviduct (containing the most developed eggs if present) and the number of vitellogenic eggs, which are the white (yolk), non-transparent and non-deformed portion of the eggs found in the ovarioles.

## Statistical analysis

For the fertility analysis we carried out individual Wilcoxon-signed rank tests for each of the measures of fertility (minimum, maximum and average ovariole width and number of vitellogenic eggs) as response variables against treatment (either methoprene or acetone control (CoA)).

For the behavioural statistical analysis, we used the programme PRIMER 6, version 6.1.13, + add-in, version 1.0.3 (PRIMER-E Ltd.) to perform permutational multivariate analysis of variance (PERMANOVA). PERMANOVA is a non-parametric MANOVA, which has the advantage that it is free from assumptions on data distributions (*Anderson, Gorley & Clarke, 2008*). All tests were carried out using 9,999 permutations on a resemblance matrix using Euclidean distance as a distance estimate. In all cases we used treatment as a fixed factor and colony as a random predictor variable to account for the structured nature of the data. Interaction between the factors was included, but removed from the final minimum adequate model when nonsignificant. All response variables were z-transformed prior to treatment in order to account for difference in units and variation between variables, which facilitates the interpretation of results in particular interactions between variables (*Gotelli & Ellison, 2004*).

To test for potential differences in weight and size between workers belonging to different colonies and treatments, both were used as response variables in a PERMANOVA. We also used PERMANOVA to investigate the effects of treatment and colony on fertility and behaviour. We used the change in behaviour following JH treatment as response variables for analysis. We calculated the mean behaviour and hierarchy position (number of winning, dominant encounters) before and after treatment to obtain a robust estimate for brood care, aggression, 'boldness', activity and sociability (ants in close proximity and distance to the nearest ant) and position in the hierarchy before and after treatment. We then calculated the change in behaviour in response to treatment by subtracting the averaged behaviour value before treatment from the average value after treatment; positive values therefore indicate an increase and negative values a decrease in response to treatment. The same calculation was performed for the change in rank (number of encounters won in dyadic interactions). To further explore the qualitative differences between the treatments, we performed a one-way similarity of percentage (SIMPER) analysis, a data exploration technique that

**Table 1  Median values for physiological traits between acetone solvent control (CoA) individuals and methoprene treated Dinosaur ant individuals.**

|  | Weight [µg] | Head width [mm] | Ovariole max width | Ovariole min width | Oovariole average width | Oocyte number |
|---|---|---|---|---|---|---|
| CoA | 335 | 4.9 | 0.186 | 0.247 | 0.208 | 0 |
| Methoprene | 374 | 4.959 | 0.1075 | 0.142 | 0.122333334 | 0 |

**Table 2  Results of the fertility Wilcoxon analyses comparing Dinosaur ant individuals treated with either acetone solvent control (CoA) or methoprene.** Presented are the *W* test statistic and *P* values for each test carried out.

|  | W | P-value |
|---|---|---|
| Oocyte number | 187.5 | 0.015 |
| Ovariole width (minimum) | 200 | 0.021 |
| Ovariole width (maximum) | 212.5 | 0.006 |
| Ovariole width (average) | 202.5 | 0.017 |

calculates the contributions individual factors make to both group (treatment) coherence and separation in a multidimensional scaling (MDS) analysis.

## RESULTS

The experimental ants did not differ in size or weight between treatments or colonies (respectively: Pseudo $F_{1,31} = 0.35$, $P = 0.72$; Pseudo $F_{11,31} = 1.04$, $P = 0.4$; Table 1), and worker fertility also did not differ between colonies (Pseudo $F_{11,31} = 1.82$, $P = 0.11$). However, worker fertility was affected by treatment, with JHa-treated individuals being less fertile compared to those treated with acetone control under all measures of fertility taken here (oocyte number, average ovariole width, minimum ovariole width and maximum ovariole width; 22; Tables 1 and 2; Fig. 1). The CoA ants were more variable in fertility than the JHa group (Fig. 1). Furthermore, there were smaller differences in fertility within the JHa group (Fig. 1). In contrast to the effects of JHa treatment on worker fertility, we found no significant differences between treatment or colonies on social dominance behaviour (respectively: Pseudo $F_{1,31} = 0.74$, $P = 0.6$; Pseudo $F_{11,31} = 1.37$, $P = 0.06$; Fig. 2).

## DISCUSSION

Our results show that JH has a role in regulating reproduction in *D. quadriceps*. Experimentally elevated JH levels not only decreased the number of vitellogenic eggs in high-ranked workers, but also resulted in an overall decrease in the size of individual ovarioles, indicating a substantial reduction in reproductive potential. In solitary insects JH often has the opposite effect by stimulating the production of vitellogenic oocytes and the same is true for many primitively eusocial insects, such as non-swarm founding wasps and bumblebees (*Robinson & Vargo, 1997*). It has been suggested that the functional reversal of JH in reproduction was important in the evolution of complex societies, and a large number of studies demonstrate often radical changes in the regulatory architecture of reproduction in eusocial species (*Robinson & Vargo, 1997; Hartfelder, 2000; Bloch et al.,*

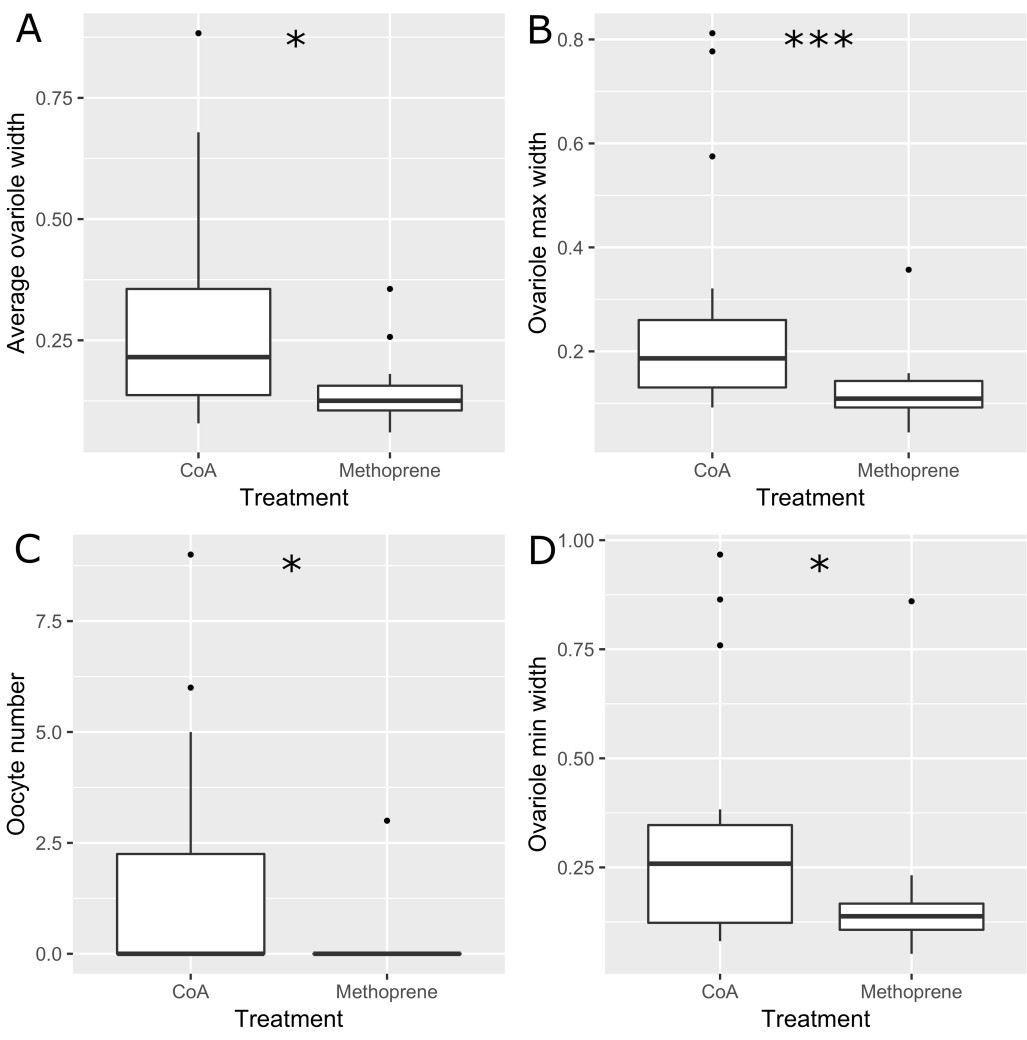

**Figure 1** **Boxplots showing fertility estimators measured in 32 *Dinoponera quadriceps* high ranked workers.** Half of the ants were treated with Juvenile Hormone analogue (Methoprene) and half with acetone control (CoA). Fertility measures were the average ovariole width (A), maximum ovariole width (B), number of oocytes (C) and minimum ovariole width (D). Stars above plots indicate significant differences following Wilcoxon tests.

*2009*). The classic example for this argument is the remodelling of the regulatory function of JH in honeybees and some ants (*Robinson & Vargo, 1997*; *Bloch et al., 2009*; *Pamminger et al., 2016*; *Azevedo et al., 2016*; *Pamminger, Treanor & Hughes, 2016*), however a small number of studies clearly indicates that high social organization is possible without it (e.g., *Brent & Vargo, 2003*; *Kelstrup, Hartfelder & Wossler, 2015*). *D. quadriceps* supports the former findings by demonstrating that a remodelling of JH function which inhibits reproduction can also be associated with simple social organization. This makes sense given the evolutionary position of *D. quadriceps*, with *Dinoponera* having secondarily reverted to simple, queenless societies from a highly eusocial ancestor (*Peeters, 1997*; *Monnin & Peeters, 1998*; *Monnin & Peeters, 1999*). This further supports the notion that there is
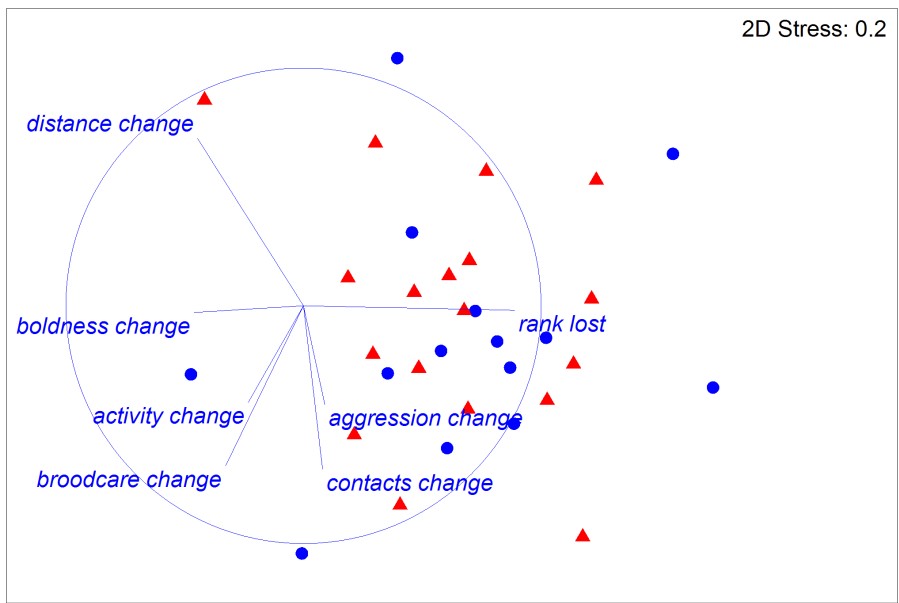

**Figure 2** **Multidimensional scaling (MDS) plot of all behaviours measured in 32 *Dinoponera quadri-ceps* high-ranked workers.** Half of the ants were treated with Juvenile Hormone analogue (JHa; red trian-gles) and half with acetone control (CoA; blue circles). Behaviours measured were brood care, sociability as distance from nearest ant and number of ant contacts, activity, 'boldness' and aggression, as well as the change in rank following treatment. Vector lines indicate the strength and contribution of the individual traits for group separation between the two treatment groups. There were no significant differences be-tween the treatments.

likely no causal link between the remodelling of the JH function in reproduction and the organisational complexity of insect societies.

The results presented here, that JH significantly decreases fertility, are therefore consistent with those for other gamergate-led ant societies (*Harpegnathos*, *Streblognathus* and *Diacamma*; *Sommer, Hölldobler & Rembold, 1993*; *Cuvillier-Hot, Lenoir & Peeters, 2004*; *Penick, Liebig & Brent, 2011*). The role of JH in such societies is thus relatively clear, but there remain questions on how JH functions in true queens of other species. Importantly in these societies JH appears to be involved as part of an 'honest' signal that informs other colony members about their fertility (*Cuvillier-Hot, Lenoir & Peeters, 2004*), a key cue for the maintenance of dominance hierarchies in these societies. In studies on a gamergate ant society, a social wasp and a termite, topical applications of JH affect cuticular hydrocarbon (CHC) profiles of adult and larval individuals such that the profile becomes more 'reproductive-like' and is perceived so by colony members (*Kelstrup et al., 2014*; *Brent et al., 2016*; *Penick & Liebig, 2017*). However in *Streblognathus* ants alphas treated with JH had their fertility reduced and the CHC profile matched more closely with that of a sterile worker than an alpha (*Cuvillier-Hot et al., 2004*)

In contrast to the relatively well-studied effects of JH on reproductive physiology, little is known about the regulatory role of JH in behaviour for most insects. The association between JH and aggression, maternal behaviour and activity has been documented in

insects (*Nijhout, 1998*; *Pearce, Huang & Breed, 2001*; *Tibbetts & Izzo, 2009*; *Tibbetts, Vernier & Jinn, 2013*), but these studies are restricted to only a handful of species. In the honey bee *Apis mellifera*, JH, in combination with the yolk precursor vitellogenin, regulates one of the major behavioural transitions in the adult honeybee worker from within-nest behaviour to external foraging (*Robinson & Vargo, 1997*). This transition is associated with a major remodelling of the behaviour repertoire and indicates the far-reaching regulatory potential of JH in behaviour. A similar function of JH has been documented in *Pogonomyrmex californicus* harvester ants and *Acromyrmex echinator* leaf-cutting ants (*Dolezal et al., 2009*; *Norman & Hughes, 2016*), demonstrating that JH can generate forager-like behavioural phenotypes. In contrast to our expectations, we find no measurable effects of JH on worker behaviour or position in the hierarchy in *D. quadriceps*. This could indicate that JH, fertility and dominance are decoupled in *D. quadriceps*. Indeed, a lack of behavioural effect of JHa on alphas in *Streblognathus* ponerine ants has been reported previously (*Cuvillier-Hot et al., 2004*). However, it is more likely that our study simply lacked the power to detect an effect. Our study was deliberately focussed on only high-ranked workers because these were the individuals in which both positive and negative effects could potentially be seen, and it may be that inclusion of ants from the full spectrum of the social hierarchy or using in-nest behavioural observations may reveal effects.

## CONCLUSIONS

Although JH is now known to have an important (though variable) role in the physiology of reproductive dominance in social insects, other hormones are also likely to be as, or more, important to reproductive status and social dominance in *D. quadriceps*. In particular, ecdysone or vitellogenin generate observed rank-specific phenotypes in other social insects (*Hartfelder, 2000*), and dopamine may also play a central role in the regulation of dominance and reproduction in species with simple societies (*Sasaki, Yamasaki & Nagao, 2007*; *Okada et al., 2015*; *Ohkawara & Aonuma, 2016*). Further work combining behavioural, genetic and physiological work is needed to illuminate the regulatory underpinning of reproductive hierarchies in simple ant societies. When looking at the broader phylogenetic picture there is accumulating evidence that JH occupies a stunning range of different, often opposite, regulatory functions. The question of how such incredible regulatory flexibility is possible without compromising fitness-relevant functions is intriguing and a promising target for further molecular and comparative investigations.

### Funding

The work was funded by the German Research Foundation (DFG) and an EC FP7 Marie Curie Fellowship PIEF-GA-2013-626585 (Tobias Pamminger), and the Biotechnology and Biological Sciences Research Council BB/J011339/1 (Victoria Norman). The funders had no role in study design, data collection and analysis, decision to publish, or preparation of the manuscript.

## Grant Disclosures

The following grant information was disclosed by the authors:

DFG (German Research Foundation).

EC FP7 Marie Curie Fellowship: PIEF-GA-2013-626585.

Biotechnology and Biological Sciences Research Council: BB/J011339/1.

## Competing Interests

The authors declare there are no competing interests.

## Author Contributions

- Victoria C. Norman conceived and designed the experiments, performed the experiments, analyzed the data, contributed reagents/materials/analysis tools, prepared figures and/or tables, authored or reviewed drafts of the paper, approved the final draft.
- Tobias Pamminger conceived and designed the experiments, performed the experiments, contributed reagents/materials/analysis tools, authored or reviewed drafts of the paper, approved the final draft.
- Fabio Nascimento contributed reagents/materials/analysis tools, approved the final draft.
- William O.H. Hughes conceived and designed the experiments, contributed reagents/materials/analysis tools, authored or reviewed drafts of the paper, approved the final draft.

## Field Study Permissions

The following information was supplied relating to field study approvals (i.e., approving body and any reference numbers):

Instituto Brasileiro do Meio Ambiente e dos Recursos Naturais (IBAMA) provided permit number 14BR004553 for this study.

## Data Availability

Raw data for Norman et al. are available as a Supplemental Figure.

## Supplemental Information

Supplemental information for this article can be found online at http://dx.doi.org/10.7717/peerj.6512#supplemental-information.

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
