# Peer review of "The role of juvenile hormone in regulating reproductive physiology and dominance in Dinoponera quadriceps ants"

_PeerJ, doi:10.7717/peerj.6512_

## Round 0.1 · original submission · Major Revisions

Dear Dr. Pamminger and colleagues:

Thanks for submitting your manuscript to PeerJ. I have now received three independent reviews of your work, and as you will see, the reviewers raised many concerns about the research. In particular, please address comments about experimental design, data collection, statistical analyses, explanation of protocols and approaches and improvements to overall presentation. Please also make sure that all terms are defined. Please address the concern by Reviewer 2 regarding the results not strongly supporting your claim that JH has an effect. Reviewer 3 also raised a similar concern, and it appears that a reanalysis is required to fully understand the effects of JH. Finally, several reviewers point out literature that seems to be missing from your cited references.

Therefore, I am recommending that you revise your manuscript accordingly, taking into account all of the issues raised by the reviewers. I look forward to seeing your revision, and thanks again for submitting your work to PeerJ.

Good luck with your revision,

-joe

Reviewer 1 ·

Basic reporting

The study by Pamminger et al. investigated the role of juvenile hormone (JH) in dominance and fertility in the queenless ant, Dinoponera quadriceps. The researchers applied a juvenile hormone analog (JHa) to dominant workers to measure changes in fertility and dominance rankbehavior. The results showed a decrease in fertility associated with JHa treatment, which is consistent with patterns in other ponerine ants; however, there were no significant effects of JHa treatment on dominance rank or behavior.

The manuscript was well written and conceived, and therefore I recommend a "pass" in this section.

Experimental design

The experiments were well designed. The sample sizes were low, but not outside norm for these types of studies. The authors report that one reason they may not have detected an effect of JHa treatment on dominance rank/behavior was due to insufficient sample size, but they do report a significant effect of JHa treatment on fertility.

The methods were explained in sufficient detail and were reasonable given constraints of the system; therefore, I recommend a "pass" for experimental design.

Validity of the findings

Statistical analyses were sound, and the results were convincing. The conclusions were also well stated, and alternative explanations of negative results were provided.

Additional comments

34-35 – Why do you think that additional samples would lead to a significant, negative relationship between JHa treatment and dominance behavior? Was there a non-significant trend? If not, this seems too speculative. Other studies have also found effects of JHa treatment on fertility but not behavior, so this result is not so surprising (e.g., Cuvillier-Hot et al. 2004).

57 – Consider starting a new paragraph beginning with, “While most modern ants have a specialized queen caste…”

72-74 – Missing citiations. Consider citing the work of Elizabeth Tibbets and Zachary Huang on paper wasps as well as the work of Sasaki on Polistes chinensis.

109-111 – It would be helpful to lay out predictions here (e.g., “If JH functions similarly to other queenless ant species, then we predict JH will suppress ovarian development and cause high ranking workers to decrease in status. If JH functions as a gonadotropin, similar to solitary insects and social wasps, then we predict…)

170 – The term “aggression” seems imprecise in this context. Two types of aggression exist in this species: defensive aggression towards an intruder, and dominance-based aggression towards another nest mate. Here, you are referring to the former, and at least mentioning this would be helpful.

196 – Is “distal” the correct term here? I’ve also seen this referred to as “proximal.” Maybe leave out this terminology and just say the “end closes to the oviduct with most fully developed eggs if present.”

250 – Should be “decreased.”

267 – Delete extra period.

Discussion – I think there is room here to place these findings in the broader context of ant reproduction, dominance, and fertility signaling. The results presented here are consistent for other gamergate led ant societies (Harpegnathos, Streblognathos, Diacamma). The role of JH therefore seems to be clear in these societies, and there remain more questions about how JH functions in true queens of other ant species. Previous studies have also investigated the connection between JH and signals of fertility. Your clear demonstration that JHa reduces fertility provides an important contribution to this area of research. I would suggest including an additional paragraph that addresses these points, including reference to the role of JH and fertility signaling. Several citations below relate to this in ants, termites, and with respect to signaling the caste of ant brood:

Cuvillier-Hot, V., Lenoir, A., & Peeters, C. (2004). Reproductive monopoly enforced by sterile police workers in a queenless ant. Behavioral Ecology, 15(6), 970-975.

Brent, C. S., Penick, C. A., Trobaugh, B., Moore, D., & Liebig, J. (2016). Induction of a reproductive-specific cuticular hydrocarbon profile by a juvenile hormone analog in the termite Zootermopsis nevadensis. Chemoecology, 26(5), 195-203.

Penick, C. A., & Liebig, J. (2017). A larval ‘princess pheromone’identifies future ant queens based on their juvenile hormone content. Animal Behaviour, 128, 33-40.

Kelstrup, H. C., Hartfelder, K., Nascimento, F. S., & Riddiford, L. M. (2014). The role of juvenile hormone in dominance behavior, reproduction and cuticular pheromone signaling in the caste-flexible epiponine wasp, Synoeca surinama. Frontiers in zoology, 11(1), 78.

284 – “Ectatomma” should be “Streblognathus” if it is consistent with the citation.

Figures – Figures 1&2 are not super intuitive and somewhat obscure the main results. I would prefer to see boxplots of ovarian traits for JHa-treated and control workers. There appears to be a good amount of overlap in Fig. 1, and a boxplot would show this clearer.

Raw data supplement – Please add explanation for “treatments 1 and 2”; also, please add units throughout (e.g., head width, weight, etc.)

Reviewer 2 ·

Basic reporting

The text is clear and well written. The experiment is well put in context and supported by adequate references.

Experimental design

It is a pity that you measured fertility based on three ovarioles only (is it per ovary or per individual?) (lines 194-199). D. quadriceps has a low fecundity and egg-laying females have few large oocytes in their ovaries hence randomly selecting three ovaries only for measurements may not provide an accurate picture. Given that D. quadriceps has 10 ovarioles per ovary only measuring all ovaries was likely feasible.

Validity of the findings

In the Results section you provide statistical results only. You do not provide values for the variables measured (e.g. actual weight, size and fertility of individual, lines 133-235). This makes for a rather disembodied paper and hinders the perception of the results and their biological significance. I would recommend adding the values of variables.

My main concern is that the results do not strongly support your claim that JH has an effect. Figure 1 shows that there is a difference between the treatment (red) and the control (blue). But it is also clear that a majority of the individuals show no difference and this is not taken into account in your interpretation. This is problematic because this is not a “classic” case where some individuals do not respond to a treatment for whatever reason, so that responding and non-responding treated individuals are split into two groups with non-responding individuals grouped with the control group. Here it is the control group that behaves strangely. The figure shows that all treated individuals responded and are grouped together (although one individual reacts slightly differently to the treatment) and that it is the control individuals that are unexpectedly split into two groups. 75% of control individuals (12/16) responded similarly to treated individuals, which clearly does not support your claim that JH treatment had an effect. In addition, the 4 control individuals responding differently (i.e. as expected from controls) are spread more than the 28 responding individuals (12 controls + 16 treatments) which is also unexpected. All this is clearly problematic and casts a serious doubt on the results.

Additional comments

Some minor comments are listed below:

Line 40-41: rephrase the sentence as “its” currently refers to “animals” rather than to “group” as intended

42: fishes instead of fish

50: eusociality does not imply monogyny and indeed many social insect species are polygynous hence please rephrase, e.g. to “complex societies in which ONE OR SEVERAL QUEENS PRODUCE all females...”

63-66. It could be added here that subordinate workers play a role in stabilizing the dominance hierarchy between top individuals in some species, including D. quadriceps (Monnin et al 2001 Nature)

88-89: Harpegnathos ants do not have queenless colonies! H. saltator colonies are founded by a queen and upon her death some workers mate (i.e. become gamergate) and carry on reproducing. The queen caste has been retained in roughly half of the species where workers can mate (e.g. Monnin & Peeters 2008 Naturwissenschaften) and its biological importance varies from anecdotic to central as in H. saltator. Please rephrase accordingly

115: quadrIceps instead of quadrAceps

237: “CoA” has not been defined yet. Please define it explicitly

281: “we find no measurable effects” would be better than “we did not find here any measurable effects”

366-368: these two references are the same

Fig. S1: move the units in the column titles (weight [mg] ; JHa [µg]) as you did for the concentration of JHa ([µg/mg])

Reviewer 3 ·

Basic reporting

The manuscript examined effects of JH on physiological state and behavior in a queenless ant using topical application of JHA. They found that some individuals treated with JHA reduced their reproductive abilities. These results valuable to understand physiological mechanisms of regulation of reproductive division of labor in ants. However, the authors should answer some points to clarify their findings. And I found that the manuscript has many typos. I strongly recommend you to check again.

Major comments
Introduction:
I imagine that you have data about correlation between JH titer and ovary development/behavior in this species. Please add the information if you have.

Statistical method:
The authors used multivariate analyses to comparisons of effect of JHA treatment on behavior and physiology. However, in this case, comparisons of effect of JHA on each response variable are more suitable to interpretation of the effect of JHA. Please carry out reanalyses to clarify the effects of JHA.

Minor comments
L. 54: Should rewrote words, “between queens and workers”, because worker-worker conflict also occur in such society.

L. 88: Okada et al., (2005) is incorrect in this context. The study focused on not JH but biogenic amines.

L. 103: Please explain “phenotypic”. Behavioral or/and physiological phenotype?

L. 123: How long observation time per a day? Also please add information about total observation time.

L. 135: Please include “is” before “observed” and “recorded”, respectively.

L. 152: Also please add information about total observation time.

L. 166: AntTrak?

L. 210 Please explain why did you use non-parametric test, although all data has been converted to z-transformation.

L. 229: non-parametric?

L. 267: Defeat a period.

L. 368: Correct??

Experimental design

no comment

Validity of the findings

no comment

Additional comments

no comment

---

## Round 0.2 · accepted · Accept

Dear Dr. Pamminger and colleagues:

Thanks for re-submitting your manuscript to PeerJ, and for addressing the concerns raised by the reviewers. I now believe that your manuscript is suitable for publication. Congratulations! There a few minor items to address, per reviewers 2 and 3. Please handle these while in production. I look forward to seeing this work in print, and I anticipate it being an important resource for the communities studying the function and physiology of juvenile hormone and ant biology. Thanks again for choosing PeerJ to publish such important work.

Good luck with your revision,

-joe

# Reviewer 2 ·

Basic reporting

Following a comment from referee 1 the authors changed figure 1 to Boxplots. This made me realised that I had misunderstood the previous version of Figure 1. My apologies for it! The new version of Figure 1 is much clearer and clarifies my previous concern about controls vs. treated individuals.

Experimental design

OK

Validity of the findings

OK

Additional comments

Some minor comments are listed below:
- Line 51: delete “a” from “in which a one or several queens”
- Line 69: the paper I was previously alluding to is Monnin et al (2002) Pretender punishment induced by chemical signalling in a queenless ant. Nature 419:61-65. Sorry I got the year wrong! It shows that subordinate workers prevent the replacement of the gamergate by a high ranking worker, because it increases their inclusive fitness. It seemed relevant given that this is the species you are experimenting with, but this is up to you
- 207 and 208: this sentence is odd. Does “the ovarioles most end closest to the oviduct” mean “the ovarioles closest to the oviduct”? Does “furthest developed eggs” mean “most developed eggs”? Please rephrase
- 221, 489 and Table 2: Wilcoxon (with capital letter) instead of wilcoxin
- 285: please rephrase this sentence, e.g. “an ‘honest’ signal THAT INFORMS other colony members about their fertility”
- 286-290: if I understand it correctly, this sentence should be “In studies on a gamergate ant society, a social wasp and a termite, topical applications...” (i.e. add “A” termite and replace – by a comma)
Table 1, legend: missing “i” in indivduals

Reviewer 3 ·

Basic reporting

Some parts of the manuscript were clarified in the current version. However, some problems are still remain. For example, although the authors had analyzed data for effects of JHA on ovary development, there are no the results of statistics in the manuscript. Also, although they were pointed out several typos in previous round, some words still unclear. Therefore, I recommend that the paper should be clarified again.

Experimental design

no comment

Validity of the findings

no comment

Additional comments

Line: 184. This software name is described as “AntTrak” (Tranter et al. 2014). However, you wrote the name, “Antrak”. Which is a correct name?
Line 193 and 263: I think that “CoA” and “coA” mean same treatment. If so, please use the same terminology throughout the manuscript. 

Line: 222-225. You should analyze the data using GLMM, and please add the results of new statics in the section RESULT.
Line: 222 and 488. “Wilcox"o"n is incorrect name.